# Anastrozole and Tamoxifen Impact on IgG Glycome Composition Dynamics in Luminal A and Luminal B Breast Cancers

**DOI:** 10.3390/antib13010009

**Published:** 2024-02-01

**Authors:** Borna Rapčan, Matko Fančović, Tea Pribić, Iva Kirac, Mihaela Gaće, Frano Vučković, Gordan Lauc

**Affiliations:** 1Faculty of Pharmacy and Biochemistry, University of Zagreb, Ante Kovačića 1, 10000 Zagreb, Croatia; glauc@genos.hr; 2Genos Ltd., Borongajska Cesta 83H, 10000 Zagreb, Croatia; mfancovic@genos.hr (M.F.); tpribic@genos.hr (T.P.); fvuckovic@genos.hr (F.V.); 3Genetic Counselling Unit, University Hospital for Tumours, Sestre Milosrdnice University Hospital Center, Ilica 197, 10000 Zagreb, Croatia; iva.kirac@kbcsm.hr (I.K.); mihaela.gace@kbcsm.hr (M.G.)

**Keywords:** breast cancer, immunoglobulin G, N-glycosylation, capillary gel electrophoresis, liquid handling

## Abstract

This study examines the intricate relationship between protein glycosylation dynamics and therapeutic responses in Luminal A and Luminal B breast cancer subtypes, focusing on anastrozole and tamoxifen impacts. The present methods inadequately monitor and forecast patient reactions to these treatments, leaving individuals vulnerable to the potential adverse effects of these medications. This research investigated glycan structural changes by following patients for up to 9 months. The protocol involved a series of automated steps including IgG isolation, protein denaturation, glycan labelling, purification, and final analysis using capillary gel electrophoresis with laser-induced fluorescence. The results suggested the significant role of glycan modifications in breast cancer progression, revealing distinctive trends in how anastrozole and tamoxifen elicit varied responses. The findings indicate anastrozole’s association with reduced sialylation and increased core fucosylation, while tamoxifen correlated with increased sialylation and decreased core fucosylation. These observations suggest potential immunomodulatory effects: anastrozole possibly reducing inflammation and tamoxifen impacting immune-mediated cytotoxicity. This study strongly emphasizes the importance of considering specific glycan traits to comprehend the dynamic mechanisms driving breast cancer progression and the effects of targeted therapies. The nuanced differences observed in glycan modifications between these two treatments underscore the necessity for further comprehensive research aimed at thoroughly evaluating the long-term implications and therapeutic efficacy for breast cancer patients.

## 1. Introduction

Protein glycosylation is a well-regulated, evolutionarily conserved type of co- and post-translational modification involving the covalent attachment of glycans. Depending on the type of linkage between oligosaccharides and proteins, we distinguish N-, O-, C-, S-, and phosphoserine glycosylation. At least half of all human proteins are estimated to be glycosylated [1,2]. Glycans are present either in free form or attached to proteins, lipids, and nucleic acids. They determine the structure and function of proteins and play a crucial role in nearly all physiological processes [2]. The most prevalent and extensively studied form of glycosylation is N-glycosylation, where the glycan attaches to the protein via the nitrogen in the side chain of asparagine.

Immunoglobulin G (IgG) is a simple glycoprotein found in human plasma. IgG contains two conserved N-glycosylation sites in the Fc region, while 15–25% of plasma IgG has attached glycans in the variable Fab regions [3]. The glycosylation of IgG has been well studied, and it is known that changes in glycosylation can affect the protein’s function. IgG serves as a key mediator of the immune response by binding to Fc gamma receptors (FcγRs) on immune effector cells [4] and by activating the C1q component of the complement, leading to complement-dependent cytotoxicity. Glycosylation determines whether IgG molecules will have pro-inflammatory or anti-inflammatory effects. Galactosylated or sialylated IgG molecules are frequently linked to anti-inflammatory effects, while their non-galactosylated or non-sialylated counterparts tend to be associated with pro-inflammatory effects [5]. Additionally, glycans lacking core fucose have been reported to display pro-inflammatory effects, potentially triggering antibody-dependent cellular cytotoxicity [5]. The glycosylation of IgG changes based on physiological conditions, few examples being ageing [6], sex hormone influence [7], and changes in lifestyle factors such as changes in body weight [8] and pathophysiological states, including rheumatoid arthritis [9], diabetes [10], cancers [11], cardiovascular diseases [12], infectious diseases [13], among others. Glycans are highly stable within individuals (intraindividual stability), and the observed changes in pathological conditions make them potential candidates as diagnostic and prognostic biomarkers.

Due to the complex structure and similar chemical and physical properties among glycans, the analysis and separation of individual glycan structures have historically been challenging, time-consuming, and labour-intensive, particularly when dealing with a large number of samples. Such manual processes are prone to human error [14]. Automating certain steps can significantly increase the number of samples processed per day, improve analysis reproducibility, reduce the potential for errors, and mitigate risks for personnel working with potentially hazardous samples [15]. Sample preparation is often the most demanding and time-consuming step in glycomics analysis, and automating this step would accelerate the analysis significantly. It is important to note that automatically prepared samples are compatible with capillary gel electrophoresis with laser-induced fluorescence, a method commonly used to analyse manually prepared samples. This technique allows for the rapid and efficient quantitative analysis of numerous glycans in a relatively short time [16].

Breast cancer (BC) is the most commonly diagnosed cancer among women [17]. Around 80% of breast tumours are oestrogen receptor (ER)-positive [18]. This is important as ER-positive breast tumours can be treated with anti-oestrogen therapy. However, this therapy is not perfect, as 60% of patients develop resistance [19]. Postoperative anti-oestrogen therapy for oestrogen-receptor-positive BC involves various treatment strategies. Moreover, beyond its postoperative usage, oestrogen therapy also serves as a valuable neo-adjuvant option, aiding in tumour reduction before surgery and contributing to improved surgical outcomes. One approach is ovarian function suppression to reduce oestrogen production (usually with gonadotropin-releasing hormone agonists). Another group of drugs inhibits aromatase, an enzyme that catalyses the final step in oestrogen biosynthesis. A third group of drugs inhibits or blocks oestrogen action by competitively binding to oestrogen receptors on BC cells, preventing oestrogen binding. These drugs are called selective oestrogen receptor modulators (SERMs). The effects of SERMs depend on the specific organ or tissue in which they bind to oestrogen receptors, acting as oestrogen agonists or antagonists, and providing selective effects. Additionally, there is a subset of drugs known as selective oestrogen receptor downregulators (SERDs), used to treat BC, especially in cases of resistance to anti-oestrogen therapy. They inhibit oestrogen receptor activity, blocking oestrogen binding and its effects. SERDs are useful for oestrogen-sensitive breast tumour growth inhibition and treating resistance to therapies like SERMs or AIs [19].

Oestrogen plays a complex role in regulating inflammatory processes [20,21], primarily through oestrogen receptors (ERα and ERβ) on immune cells and through direct effects on the cell nucleus [22]. Changes in oestrogen and progesterone levels can influence disease activity [23]. High oestrogen levels during pregnancy have been shown to improve symptoms and alleviate inflammatory and autoimmune diseases in women [9]. Improvement in autoimmune and inflammatory disease symptoms during pregnancy correlates with a shift from a pro-inflammatory glycan profile of IgG with high levels of agalactosylated glycoforms towards an increase in galactosylated, anti-inflammatory glycoforms [24]. The drop in oestrogen levels after childbirth and return to physiological levels often leads to disease reactivation. A similar effect has been observed during different phases of the menstrual cycle [25]. In a study using a mouse model, researchers discovered that high oestrogen levels stimulate the expression of the enzyme ST6GAL1, which increases the proportion of anti-inflammatory, sialylated IgG glycoforms [26]. It was confirmed that oestrogen affects IgG glycosylation by increasing digalactosylation trait [7].

The literature on IgG glycosylation in BC is scarce; previous research has primarily focused on changes in overall protein glycosylation in serum. However, Kawaguchi-Sakita et al. demonstrated significant differences in Fc IgG glycosylation between women with BC and healthy controls, and they developed a model for predicting BC using IgG, with an AUC value of 0.874 [27]. The significance of specific glycan structures, including FA2, as potential biomarkers for early detection of BC, was earlier underscored in the work by Gebrehiwot et al., which reported elevated levels of specific agalactosylated glycan structures (FA2 and FA2B) in BC patients [28].

To date, there are no studies that have tracked changes in IgG glycosylation as a response to anti-oestrogen therapy in BC patients. Given the established influence of BC and oestrogen on IgG glycosylation, it can be expected that such therapy will impact IgG glycosylation.

## 2. Materials and Methods

### 2.1. Participants

This study involved adult individuals recently diagnosed with breast cancer, utilizing MRI, ultrasound, and mammography alongside core biopsies to ascertain the clinical stage (cT1-4N+) and biological characteristics of the disease. Participants with Luminal A and Luminal B types, without evidence of metastatic disease, were included, while those undergoing neoadjuvant therapy and treatment with GnRH agonists were excluded from the sample. Menopausal status was determined for the group receiving anastrozole treatment. The study focused on patients who exclusively received adjuvant hormonal therapy following the surgical procedure.

Samples were collected at the Cancer Clinic of Sisters of Charity University Hospital Centre in Zagreb, Croatia, and patients’ IgG N-glycosylation profiles were determined. The research was approved by the Ethics Committee of the Sisters of Charity University Hospital Centre (approval number 251-29-11-20-01-2) and by the Ethics Committee of Faculty of Pharmacy and Biochemistry (approval number 251-62-03-23-59), and it was conducted in accordance with the principles of the Helsinki Declaration.

The initial biological sample was the blood plasma of the subjects, separated from blood cells from 6 mL of blood. Blood was collected by trained personnel using routine laboratory venipuncture at the Sisters of Charity University Hospital Centre. Over a span of 9 months, around 120 plasma samples were gathered at approximately 3-month intervals. These samples were obtained during surgical procedures, prior to therapy initiation, and during routine participant follow-ups; then, they were stored at −80 °C until subsequent analysis.

All samples were assigned specific encoded numbers to safeguard participant identities. Each coding number contains systematically organized data encompassing various categories. The data associated with each number comprise general information (such as the year of birth), pathohistological details concerning cancer (including disease diagnosis, history of previous medical interventions, tumour histological type, tumour size, molecular subtype, tumour, lymph nodes, metastasis (TNM) staging, histological grade, nuclear grade, axillary lymph node status, oestrogen receptor percentage (ER%), progesterone receptor percentage (PR%), human epidermal growth factor receptor 2 (HER2/neu), Silver In Situ Hybridization (SISH), and Ki-67%), information regarding the administered therapies, and specific details linked to sample collection.

### 2.2. CGE

The protocol employed in this study started with the isolation of IgG on the CIM^®^ r-Protein G LLD 0.05 mL Monolithic 96-well Plate (2 µm channels) (BIA Separations, Ajdovščina, Slovenia, Cat No. 120.1012-2) followed by glycan deglycosylation and APTS labelling, so the samples could be analysed on the CGE. This method, described by Hanić et al. [29], was a development of previous methods described by Pučić et al. [30] and adjusted by Trbojević-Akmačić et al. [31] and was adapted for the Tecan Freedom Evo automated platform. To isolate IgG, 25 µL of both plasma samples and corresponding plasma standards were diluted with 1× PBS buffer at a ratio of 1:7. Following dilution, the samples were resuspended and transferred to a vacuum filtration setup using a wwPTFE (Pall Corporation, New York, NY, USA) filter plate. Subsequently, the filtered samples were transferred to a protein G plate, where they underwent binding and consecutive washing steps with 1× PBS buffer containing 0.25 M NaCl. The bound IgG was eluted using a solution of 0.1 M formic acid neutralized with ammonium bicarbonate buffer. The eluted IgG fraction (20 µL) was then dried and made ready for subsequent protocol steps.

For the dried IgG samples, treatment involved using a solution consisting of 1.66× PBS, 0.5% SDS, 2% Igepal (Sigma-Aldrich, St. Louis, MO, USA), and 5 µL of mixture PNGase F (Promega, Madison, WI, USA), 0.12 µL per sample, mixed with 1.8× PBS to denature the IgG and release its glycans. Following the release, the glycans were labelled with the labelling mixture. The labelling mixture was freshly prepared by combining 2 μL of 30 mM APTS (Synchem, Felsberg, Germany) in 3.6 M citric acid (Sigma-Aldrich, St. Louis, MO, USA) with the 2 μL of 1.2 M 2-picoline borane in DMSO (Sigma-Aldrich, St. Louis, MO, USA) per sample and allowed to incubate for 16 h at 37 °C. The reaction was stopped by introducing 80% ACN (Carlo Erba, Milan, Italy). A Bio-Gel P-10 slurry was placed in a filter plate, connected to a vacuum filtration apparatus, and used to rinse the dry Bio-Gel P-10 residues in each well with ultrapure water and 80% ACN (Carlo Erba, Milan, Italy). The sample volume, including cold ACN, was transferred to the filter plate containing the Bio-Gel P-10. Vacuum pressure facilitated liquid passage through the filter, while purification of glycans attached to the Bio-Gel P-10 involved multiple washes with 80% ACN/100 mM TEA (Carlo Erba, Milan, Italy/MilliporeSigma, Burlington, MA, USA) to neutralize excess dye, followed by additional washes with 80% ACN (Carlo Erba, Milan, Italy) to eliminate TEA and reduce salt content. The elution of labelled glycans was accomplished using 500 µL of ultrapure water on a clean square well, round-bottom 1.2 mL collection plate (Thermo-Fischer Scientific, Waltham, MA, USA).

Finally, the purified glycans were mixed with 7 µL of Hi-Di Formamide in a MicroAmp Optical 96-Well Plate and subsequently analysed using a 3500 Genetic Analyzer (Thermo-Fischer Scientific, Waltham, MA, USA). For CGE-LIF analysis, 3 µL of purified IgG N-glycans was combined with 7 µL of Hi-Di Formamide analysed using an ABI3500 Genetic Analyzer (Thermo-Fischer Scientific, Waltham, MA, USA) equipped with 50 cm long 8-capillary array filled with POP-7 polymer (Thermo-Fischer Scientific, Waltham, MA, USA) as a separation matrix. Run parameters were set as follows: run time—1000 s, injection time—9 s, injection voltage—15 kV, run voltage—15 kV, oven temperature—60 °C. The resulting electropherograms were manually integrated into 27 glycan peaks (Appendix A) using the Empower 3 software (Waters, Milford, MA, USA). The amount of glycan structures in a peak was calculated as previously described [32] and expressed as a percentage of the total integrated area (total area normalization) and subjected to statistical analysis. From these peaks, 6 derived glycan traits were calculated using the formulas detailed in Appendix A.

### 2.3. Statistical Analysis

The CGE data outputs for glycan were initially standardized and adjusted for experimental differences to facilitate the comparison of samples. Normalization of the area under the peaks was achieved through total area normalization, followed by log transformation and batch correction using the ComBat method, as implemented in the “sva” package [33] in the statistical software R (v. 4.2.3., Vienna, Austria) [34]. Subsequently, the glycan peak values were converted back to their original scale before calculating derived traits based on the formulas provided in the supplementary table (Appendix A). To attain values with a mean of 0 and a standard deviation of 1, the glycan trait values were transformed using an inverse rank transformation to establish normality. The effect of time on the derived glycan traits was examined using a linear mixed model, with time treated as a fixed effect. The resulting *p*-values were subsequently adjusted using the Benjamini–Hochberg correction method.

## 3. Results

### 3.1. Study Overview and Inclusion Criteria

The present study examined a cohort of 40 women diagnosed with Luminal A and Luminal B subtypes of BC, without evidence of metastatic disease at the time of recruitment. Participants were selected based on their first-time diagnosis of BC, verified through MRI, ultrasound, and mammography, along with wide-core needle breast biopsy to determine the clinical stage. Exclusion criteria included prior neoadjuvant therapy and treatment with Gonadotropin-Releasing Hormone (GnRH) agonists.

Of the 40 participants, 19 participants underwent glycan analysis at three specific time points, including the baseline and 3- and 6–9-month intervals, while 15 individuals were analysed at the baseline and 1 additional time point. Additionally, 6 participants were assessed at the baseline and 3-month time points.

The study aims to provide a comprehensive understanding of the impact of specific BC therapies, focusing on tamoxifen and anastrozole therapies, on the glycosylation patterns associated with BC progression. A detailed analysis of the glycan traits, including agalactosylation (G0), monogalactosylation (G1), digalactosylation (G2), sialylation (S), bisecting GlcNAc (B), and core fucosylation (CF), is presented, offering critical insights into the dynamic nature of glycan alterations in response to these distinct therapeutic approaches.

### 3.2. Glycan-Derived Traits and Their Response to Therapy

To assess the impact of the specific BC therapies, we conducted a comprehensive analysis of their time effects on individual glycan traits. Our initial focus was on anastrozole, a representative Aromatase inhibitor. Notably, our investigation of the effects of anastrozole on glycan traits revealed distinctive trends. We observed marginal yet consistent positive incremental changes in G0 (time effect: 0.04639), B (time effect: 0.01696), and CF (time effect: 0.02219), while G1 (time effect: −0.00395), G2 (time effect: −0.03684), and S (time effect: −0.05190) displayed decreasing trends over time (Figure 1). A subsequent calculation of the *p*-values highlighted the significance of changes in G0 and S. However, following the application of the Benjamin-Hochberg correction (Table 1), the significance of both traits was no longer maintained.

Conversely, our examination of tamoxifen, a representative of SERM drugs, revealed distinct trends differing from those observed in the anastrozole treatment. Notably, all glycan traits, except for B and G2, exhibited opposing trends compared to the anastrozole therapy. Specifically, G0 (time effect: −0.02304), G2 (time effect: −0.00408), and CF (time effect: −0.05364) exhibited decreasing trends, while G1 (time effect: 0.01694), S (time effect: 0.0794), and B (time effect: 0.04254) displayed increasing trends over time (Figure 1). Unlike the anastrozole therapy, the initial analysis of tamoxifen demonstrated significant time effects for S and CF. However, akin to the findings in the anastrozole therapy, both traits proved insignificant following the Benjamin–Hochberg correction (Table 1).

### 3.3. Tamoxifen and Anastrozole Therapy Comparison

A comparative analysis between the effects of tamoxifen and anastrozole therapies on the glycan traits revealed distinct patterns, as outlined in Table 2.

The comparison highlighted notable differences in the response of specific glycan traits to tamoxifen and anastrozole therapies. While G0 exhibited a decrease (time effect of −0.05118) in response to tamoxifen compared to anastrozole, other galactosylation-related traits, G1 and G2, displayed insignificant differences between the two therapies, with positive time effects of 0.01751 and 0.02465, respectively. These effects were not found to be statistically significant both before and after the aforementioned correction.

Notably, the S trait exhibited a substantial and statistically significant, both before and after the correction, increase (time effect of 0.10811) in response to tamoxifen compared to anastrozole, suggesting a differential impact on sialylation patterns. Conversely, the B trait showed no significant changes, with a comparative time effect of 0.03973. However, CF demonstrated a marked decrease (time effect of −0.09636) with tamoxifen compared to anastrozole, indicating distinct effects on CF. Notably, this difference remained statistically significant even after the correction, underscoring the robustness of the observed disparity in CF between the two therapeutic approaches.

## 4. Discussion

This study delved into the intricate variations in glycan structures associated with different therapeutic interventions, specifically investigating the effects of Aromatase Inhibitor (AI) therapy utilizing anastrozole and SERM therapies employing tamoxifen, in Luminal A and Luminal B subtypes of BC. There are two key factors at play within this dynamic interplay.

The first important dynamic is the one of BC. The study highlights the crucial role of glycan alterations in BC progression, investigating BC glycosylation alterations beyond changes in overall serum protein glycosylation [35]. Previously, the significance of specific glycan structures, including FA2, as potential biomarkers for early detection of BC, was underscored in the work of Gebrehiwot et al., which reported elevated levels of specific G0, glycan structures (FA2 and FA2B) in BC patients [27,28]. Nonetheless, these studies did not assess the alterations occurring in glycosylation patterns during BC therapy and the corresponding responses of patients to such treatments.

The other important dynamic concerns oestrogen and its effects on IgG glycosylation. Recent research has shed light on the interconnection between oestrogen and IgG Fc glycosylation, revealing the direct regulatory role of female sex hormones in modulating IgG Fc glycosylation [26,27]. These studies elucidated the impact of oestrogens, particularly oestradiol, on IgG Fc galactosylation and sialylation, resulting in what is nominally considered a shift towards less inflammatory IgG N-glycosylation profiles, which is vital for understanding the presented results. This was also indicated by observing pregnant women who exhibited increases in G2 and S traits during pregnancy, peaking during the third trimester [36]. This effect is followed by a reduction in autoimmune pathologies in pregnant women [24]. To delve into oestrogen’s impact on IgG glycosylation, Ercan et al. had subjects initially receive a GnRH agonist to suppress testosterone production. Later, they were administered replacement transdermal testosterone, a placebo, or transdermal testosterone with anastrozole, an aromatase inhibitor. Over 12 weeks, those on testosterone showed no shift in IgG glycans, while the placebo group displayed a marked increase in G0F glycans. Intriguingly, individuals given testosterone with anastrozole exhibited a similar G0F increment as the placebo, hinting at oestrogen’s influence on IgG Fc galactosylation in males, albeit to a lesser degree due to lower oestrogen levels. Notably, testosterone alone did not directly impact IgG Fc glycosylation [7].

The changes in glycan modifications resulting from diverse therapeutic approaches, particularly in the case of anti-oestrogen treatments like anastrozole, signify the intricate interplay of multiple biological pathways. Anastrozole showed reduced levels of digalactosylation and sialylation. The rise in identified agalactosylated structures, though linked to BC, is also connected to oestrogen suppression, indicating the strong impact of suppressing oestrogen during the adjuvant therapy. This could be attributed to the distinct mechanisms of anastrozole, which broadly reduces oestrogen levels throughout the body by inhibiting aromatase, in contrast to the more targeted approach of tamoxifen, which focuses on blocking oestrogen receptors in cancer cells and did not exhibit similar patterns, showing a tendency towards a reduction in agalactosylation and increase in sialylation.

The delicate balance between the effects of anti-oestrogen therapies on specific glycan traits and the changes caused by BC influencing glycan alterations underscores the complexity of BC progression [26,27]. The comparison between the two therapies sheds more light on the interaction of these pathways, helping reveal the intricate dynamics at play.

The different effects of tamoxifen and anastrozole therapies on glycan traits emphasize the subtle variations in their therapeutic mechanisms and their respective impacts on BC advancement. A direct comparison between the two therapies revealed significant differences in glycan trait trends over the therapy period. The observed trends are particularly the notable increase in S and decrease in CF during tamoxifen therapy compared to its anastrozole counterpart. An increase in sialylation and a decrease in core fucosylation may signify a shift in the immune response and modulation of antibody functionality. Enhanced sialylation often promotes anti-inflammatory effects, contributing to the dampening of immune responses and reduction in pro-inflammatory reactions. Conversely, reduced core fucosylation can lead to increased antibody-dependent cellular cytotoxicity (ADCC) and phagocytosis, potentially indicating an upregulation of immune-mediated cytotoxic responses. Therefore, these opposing changes in sialylation and core fucosylation suggest a complex interplay between immunomodulatory processes, potentially resulting in a fine-tuned balance between inflammatory and cytotoxic responses in the immune system in response to an adjuvant BC therapy [37,38].

It is known that different individuals respond differently to this kind of therapy [39]. We have noticed several individuals who have shown a large difference compared to other measured patients. In examining the individual responses within our sample set to tamoxifen and anastrozole therapies, noteworthy outliers emerged, particularly in G0 IgG glycosylation. Among those undergoing anastrozole therapy, three individuals displayed a significant increase of 20% or more in G0 IgG glycosylation from the onset of treatment. Conversely, a solitary participant exhibited a substantial reduction of −15% in this marker. Meanwhile, within the tamoxifen group, five individuals demonstrated a notable increase of 15% or more in G0 IgG glycosylation, whereas four individuals exhibited a reduction of 15%. In the G2 subset, two individuals on anastrozole displayed an increase of 15% or more, while four individuals on tamoxifen showed similar increments. Interestingly, there were instances of inverse responses, such as two cases where anastrozole led to a 15% reduction and four cases where tamoxifen elicited the same reduction. Moreover, in one case, anastrozole S resulted in a 15% increase, while two showed a large decrease, two individuals on tamoxifen exhibited a 15% or more increase, and one individual displayed a 15% reduction while on tamoxifen. The majority of the population showed no such distinct trends, especially for G1 and CF where these trends remained more or less stable. These diverse and sometimes contrasting responses underscore the intricate variability in individual reactions to these therapies, necessitating further exploration to elucidate the underlying mechanisms governing such disparate outcomes.

However, it is imperative to recognize the limitations inherent in our study. The relatively short follow-up duration, spanning a maximum of 9 months of adjuvant therapy—despite the potential for such therapy to continue for years—may have limited our ability to capture robust and enduring trends in treatment response and patient outcomes. This constrained timeframe might not fully encapsulate the long-term effects or fluctuations that could emerge over an extended treatment period. Additionally, the absence of control groups, whether comprising healthy individuals or those receiving a placebo, is a notable limitation. The inclusion of such control groups would have provided a crucial benchmark for understanding the observed alterations in comparison to baseline or untreated conditions. This absence diminishes the depth of insight into the observed changes within the context of the studied interventions. Looking forward, future investigations should aim to extend the sampling period significantly. Doing so would offer a more comprehensive and nuanced understanding of therapeutic efficacy over prolonged durations. This extension in sampling duration would facilitate a more thorough evaluation of treatment response and outcomes, accounting for potential variations and trends that may manifest over extended periods, thereby enabling more informative comparisons and robust conclusions.

## 5. Conclusions

In conclusion, this study sheds light on the intricate interplay between glycan alterations and therapeutic interventions in Luminal A and Luminal B subtypes of BC, particularly focusing on the effects of anastrozole and tamoxifen. Our findings stress the role of glycan dynamics in BC progression, emphasizing the need for targeted investigations beyond overall serum protein glycosylation changes.

The observed complexities in glycan modifications, influenced by the delicate balance between therapeutic effects and endogenous regulatory mechanisms, highlight the multifaceted nature of BC advancement. The distinct impacts of anastrozole and tamoxifen therapies on glycan characteristics further underline the nuanced differences in their mechanisms and effects on BC progression.

While our findings suggest potential increased inflammatory processes associated with anastrozole therapy, this problem calls for further research to comprehensively evaluate the long-term trends and therapeutic efficacy. Future investigations with extended sampling periods are necessary to unravel the intricate mechanisms underlying glycan alterations and their implications in BC treatment.

## Figures and Tables

**Figure 1 antibodies-13-00009-f001:**
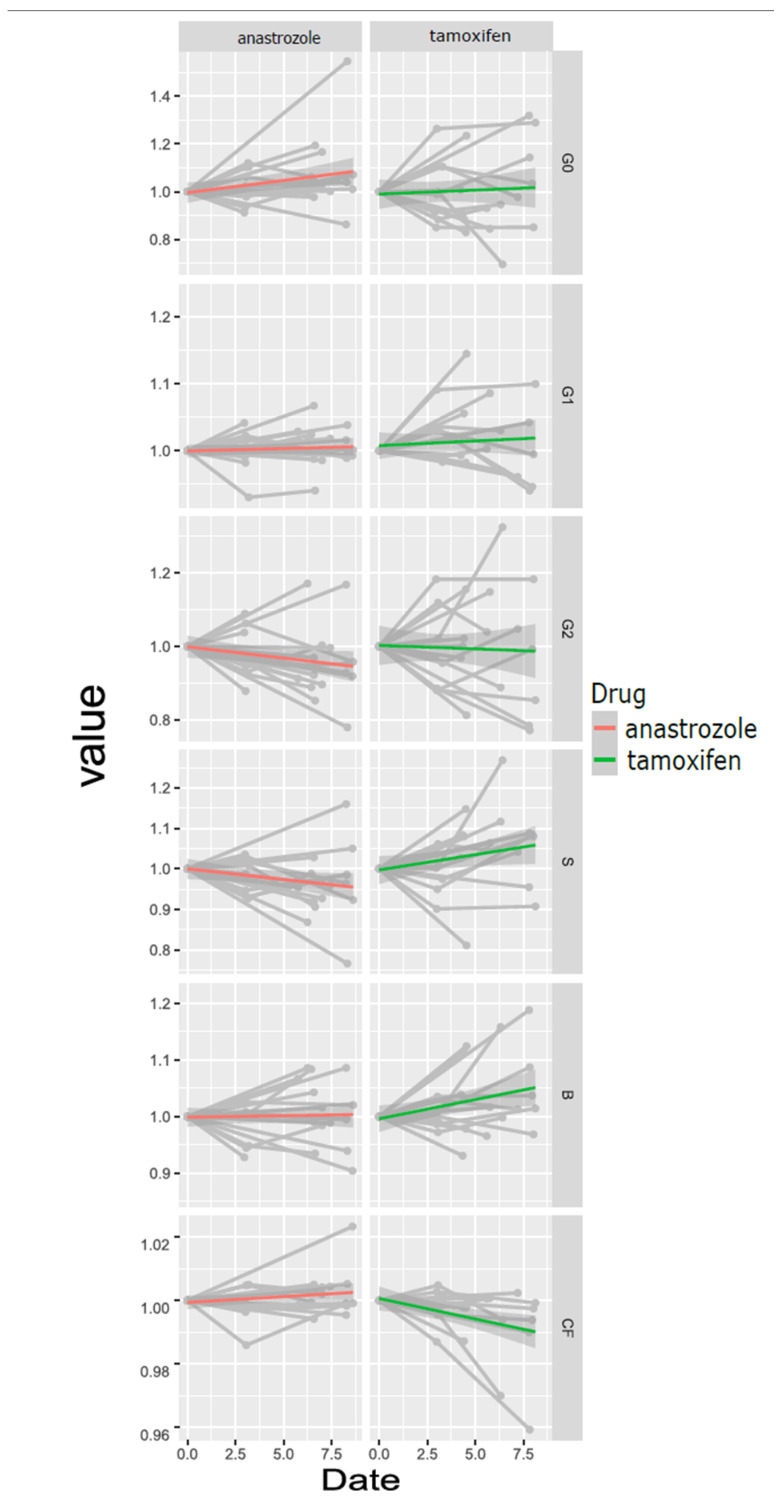
Graph representing the time effect for each glycan over the sampling period in months. The y-axis represents the normalized value calculated by dividing each subsequent value by the initial t0 value. The x-axis represents the time since the first measurement in months. The grey lines depict the participants individual changes in glycosylation trait abundance between each timepoint. The grey spots correspond to individual timepoints within the dataset. The red and green lines illustrate the overall trend observed in the analysed group of samples for anastrozole and tamoxifen respectively.

**Table 1 antibodies-13-00009-t001:** Table representing the time effect for each glycan trait and drug, along with their corresponding standard error and *p* values. The corresponding letters (G0, G1, G2, S, B, CF) represent distinct glycosylation traits: G0 for agalactosylation, G1 for monogalactosylation, G2 for digalactosylation, S for sialylation, B for bisecting N-acetylglucosamine, and CF for core fucosylation in protein structures.

Drug	Glycan Trait	Time Effect	Time Standard Error	Time *p* Value	Time-Adjusted *p* Value
anastrozole	G0	0.04639	0.01984	0.02607	0.11733
	G1	−0.00395	0.02366	0.86762	0.89407
	G2	−0.03684	0.01685	0.05286	0.19030
	S	−0.05190	0.02144	0.02243	0.11733
	B	0.01696	0.02118	0.42564	0.58935
	CF	0.02219	0.02244	0.34082	0.58935
tamoxifen	G0	−0.02304	0.02703	0.39654	0.58935
	G1	0.01694	0.03299	0.61028	0.70917
	G2	−0.00408	0.03056	0.89407	0.89407
	S	0.07940	0.02723	0.01569	0.11733
	B	0.04254	0.03125	0.19335	0.43504
	CF	−0.05364	0.01741	0.00678	0.11733

**Table 2 antibodies-13-00009-t002:** Table representing the time effect in drug comparison across each glycan trait, along with their corresponding standard error and *p* values. The corresponding letters (G0, G1, G2, S, B, CF) represent distinct glycosylation traits: G0 for agalactosylation, G1 for monogalactosylation, G2 for digalactosylation, S for sialylation, B for bisecting N-acetylglucosamine, and CF for core fucosylation in protein structures.

Drug Comparison	Glycan Trait	Time Effect	Time Standard Error	Time *p* Value	Time-Adjusted *p* Value
tamoxifen-anastrozole	G0	−0.05118	0.02691	0.06118	0.22026
	G1	0.01751	0.04121	0.67122	0.75512
	G2	0.02465	0.02923	0.40152	0.51624
	S	0.10811	0.02992	0.00083	0.01495
	B	0.03973	0.03354	0.24650	0.44371
	CF	−0.09636	0.03012	0.00301	0.02713

## Data Availability

Data are contained within the article and Appendix A.

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
