# Peer review of "Anastrozole and Tamoxifen Impact on IgG Glycome Composition Dynamics in Luminal A and Luminal B Breast Cancers"

_2073-4468, 2024, doi:10.3390/antib13010009_

Round 1

Reviewer 1 Report

Comments and Suggestions for Authors

Thank you for an important and clinically significant study. I have attached a document with my inputs. 

Title

a) Need to be shortened. b) Consider leaving out “Evaluating the Impact of”, “Therapies” and “Subtypes”. c) Write ‘c’ of “composition” in capital letter for consistency. d) Suggestion: Anastrozole and Tamoxifen on IgG Glycome Composition Dynamics in Luminal A and Luminal B Breast Cancer’.

Abstract

a) It reads like a conclusion.

b) Please revise and include at least 2 statements that necessitated the study.

Key words

a) Replace the abbreviations.

b) Otherwise, no other concerns. 

Introduction

a) Multiple brackets in line 54-57 are confusing. Please remove brackets around anything else except references and at first use of abbreviations.

b) The above i.e. removal of brackets, should be done across the entire introduction.

c) Oestrogen therapy is not only used post-operatively but also as neo-adjuvant or palliation. One would not only mention postoperative treatment (Line 76-77).

d) Otherwise: No other concerns.

Materials and methods

a) Check and revise grammar in line 119.

b) Does it mean patients with locally advanced breast, axillary disease or aggressive biology like HR+, HER2+ and high Ki67 had surgery without neo-adjuvant treatment? It would be unusual.

c) Selection of SERM or AI depends on the menopausal status. How was the menopausal state factored into the study? It would not be avisable to use AI in women who are pre-menopausal. How did the age of the Aristrozole and Tamoxifen group compare?

d) Leave it as just Luminal B and Luminal A. Preferably the writing should start with Luminal A. Luminal B (ER) and Luminal A (cT1-4N+) may confuse readers and suggest there is a subtype of Luminal B that it is not oestrogen receptor positive. Furthermore, it implies ER positivity only applies to Luminal B and not Luminal A whereas both are ER positive.

e) Abbreviations should be defined at first use. Examples include ER, PR, HER2/neu, SISH.

f) Data analysis: No issue.

Results

a) What is the difference between assessment at baseline and 1 additional point in 17 versus baseline and 3- months time points in 6 of the participants? (Line 213-215). Are both not assessment done at twice: baseline and once-off in 3-9 months?

b) Tamoxifen is not an aromatase inhibitor (AI) in Line 217.

c) Similarly, Anastrozole is not a SERM line 217-218 (see Line 224-225 that is correct).

d) G2 is not defined (Line 219). e) Correct “Tamoxifene” which should be ‘Tamoxifen’ in Figure 1.

f) Provide additional information in the legend of Figure 1 to explain what is happening. It should be comprehensive and self-explanatory. Also add a footnote to explain of the abbreviations.

g) Need to confirm that the Y-axis is indeed representing months. The period of study should ideally be in the X-axis. Normalized values should be in the Y-axis. The study was for a period of up to months, which should be in the chart.

h) Table 1: No concerns but add explanation of the abbreviations.

i) Table 2: No concerns. Need to add explanation of the abbreviations. Otherwise: No other issues.

Discussion

a) Limitations

i. Add small sample size as a limitation. There would have been multiple subsets based on age, clinical stage and receptor and Ki-67 index. 42 might not have been enough. Whether participants were pre- or post-menopausal is also relevant. Only 19 participants were studied until the end of the study 9 months.

ii. Also add heterogeneity of breast cancer and treatment response among patients.

b) Otherwise, no other concerns.

Conclusion

a) Use either Anastrozole/anastrozole consistently.

b) Otherwise, no other concerns.

References: No concerns. 

Comments on the Quality of English Language

Some editing is required.  

Author Response

We appreciate your valuable feedback and have made revisions accordingly. We shared the edited text alongside our response. All the changes were highlighted. Please review the updated content and our explanations to ensure that our adjustments align with your suggestions. Please see attached responses to the review.

Reviewer 2 Report

Comments and Suggestions for Authors

Hi,

The idea of the study is novel; however, the results need to be improved. Here are my few comments:

1. Without a control group, it is very difficult for me to compare the results.

2. The sample size of the study is very small, it would be better if the sample size is increased.

3. The introduction part should be more focused on the title of the study. I found the reference 24-28 (line no 99-112) are not much related to the introduction part.

4. I feel some more results need to be incorporated to give solidarity to your findings.

5. The discussion part lacks the research articles to correlate with your findings.

6. Anastrozole and Tamoxifen should be written uniformly as ‘anastrozole and tamoxifen’ throughout the manuscript.

Comments on the Quality of English Language

English needs to be improved, I found the grammar mistakes at few places places.

Author Response

We appreciate your valuable feedback and have made revisions accordingly. We shared the edited text alongside our response. All the changes were highlighted. Please review the updated content and our explanations to ensure that our adjustments align with your suggestions.

Reviewer 3 Report

Comments and Suggestions for Authors

“Evaluating the Impact of Anastrozole and Tamoxifen Therapies on IgG Glycome composition Dynamics in Luminal A and Luminal B Subtypes of Breast Cancer” by Borna Rapčan, Matko Fančović, Tea Pribić, Iva Kirac, Mihaela Gaće, Frano Vučković, Gordan Lauc. (Manuscript ID: antibodies-2781906)

This manuscript described a study evaluating the impact of two types of breast cancer therapies (Anastrozole and Tamoxifen) on IgG glycan compositions in estrogen receptor (ER)-positive patients (Luminal A and B subtypes). Anastrozole is an inhibitor for aromatase-an enzyme that catalyzes the final step in the estrogen biosynthesis, whereas Tamoxifen is an antagonist that competitively binds to the estrogen receptors and prevents estrogen binding to these receptors. With a cohort of 42 participants, the study evaluated the alternations in glycan structures of serum IgGs over a period of 9 months, by utilizing a series of automated processing protocols which include IgG isolation, denaturation, labeling, purification, and capillary gel electrophoresis with laser-induced fluorescence. The study uncovered distinct trends of serum IgG glycans in response to the therapies. Anastrozole therapy led to reductions in sialylation and increase in core fucosylation, whereas Tamoxifen appeared to increase sialylation and decrease core fucosylation. The manuscript therefore concluded that the district impacts of Anastrozole and Tamoxifen treatments on serum IgG glycan characteristics underlined the differences in their therapeutic mechanisms of action and the effects on breast cancer progression.

The topic of the manuscript is potentially interesting and therapeutically relevant. However the manuscript needs to address a number of major issues:

1)     First of all, the authors need to show the representative data on the key experimental results, such as serum IgG purification, and capillary gel electrophoresis with laser-induced fluorescence. The key is that the authors need to show readers how Figure 1 is generated, regarding glycan species determination. This is critical for supporting the conclusion of the manuscript.

2)     Proper references are needed for Page 1 Line 36-37 for N-, O-, C-and P-glycosylation. What is P-glycosylation?

3)     Proper references are needed for Page 2 line 50-52 for anti-inflammatory effects of galactosylated or sialylated IgG molecules, as well as pro-inflammatory effects of non-galactosylated or non-sialylated forms.

4)     In Page 3 line106-107 Kawaguchi-Sakita et al should be reference 28, not 27 (line109)? Also Line 111, Gebrehiwot et al seems to be reference 37, not reference 28. The authors should check all the references in the manuscript thoroughly.

5)     In Figure 1 (Page 6),  please show cartoon drawings of glycans for G0, G1, G2, bisecting,  core fucosylation, sialylation.

6)     In Heading section, the page numbers are incorrect throughout the manuscript.

Author Response

We appreciate your valuable feedback and have made revisions accordingly. We are sharing the edited text alongside our responses to address your comments. Please review the updated content and our explanations to ensure that our adjustments align with your suggestions.

Round 2

Reviewer 1 Report

Comments and Suggestions for Authors

Thank you very much for the revisions. Except for 40 versus 42, in Line 216 and 222, I have no further concerns. The total number of the participants should be 40 (19 + 15 + 6).  Please accept my sincere apology as I missed it (40 versus 42) during the first review. 

Comments on the Quality of English Language

The English language is significantly improved. Some of the sentences in the abstract may be simplified by leaving some of the words that not adding value. Otherwise, one has no other concerns.   

Reviewer 2 Report

Comments and Suggestions for Authors

Thank you for addressing all my concerns.